# Diagnosis Methods for COVID-19: A Systematic Review

**DOI:** 10.3390/mi13081349

**Published:** 2022-08-19

**Authors:** Renata Maia, Violeta Carvalho, Bernardo Faria, Inês Miranda, Susana Catarino, Senhorinha Teixeira, Rui Lima, Graça Minas, João Ribeiro

**Affiliations:** 1Microelectromechanical Systems Research Unit (CMEMS-UMinho), School of Engineering, Campus de Azurém, University of Minho, Guimarães, Portugal; 2LABBELS-Associate Laboratory, Braga/Guimarães, Portugal; 3MEtRICs, Mechanical Engineering Department, Campus de Azurém, University of Minho, 4800-058 Guimarães, Portugal; 4ALGORITMI, Production and Systems Department, School of Engineering, Campus de Azurém, University of Minho, 4800-058 Guimarães, Portugal; 5CEFT, Faculty of Engineering, University of Porto, 4200-465 Porto, Portugal; 6ALiCE, Faculty of Engineering, University of Porto, 4200-465 Porto, Portugal; 7Campus de Santa Apolónia, Instituto Politécnico de Bragança, 5300-253 Bragança, Portugal; 8Centro de Investigação de Montanha (CIMO), Campus de Santa Apolónia, Instituto Politécnico de Bragança, 5300-253 Bragança, Portugal; 9Laboratório Associado para a Sustentabilidade e Tecnologia em Regiões de Montanha (SusTEC), Campus de Santa Apolónia, Instituto Politécnico de Bragança, 5300-253 Bragança, Portugal

**Keywords:** COVID-19, diagnosis, image analysis, PCR, SARS-CoV-2

## Abstract

At the end of 2019, the coronavirus appeared and spread extremely rapidly, causing millions of infections and deaths worldwide, and becoming a global pandemic. For this reason, it became urgent and essential to find adequate tests for an accurate and fast diagnosis of this disease. In the present study, a systematic review was performed in order to provide an overview of the COVID-19 diagnosis methods and tests already available, as well as their evolution in recent months. For this purpose, the Science Direct, PubMed, and Scopus databases were used to collect the data and three authors independently screened the references, extracted the main information, and assessed the quality of the included studies. After the analysis of the collected data, 34 studies reporting new methods to diagnose COVID-19 were selected. Although RT-PCR is the gold-standard method for COVID-19 diagnosis, it cannot fulfill all the requirements of this pandemic, being limited by the need for highly specialized equipment and personnel to perform the assays, as well as the long time to get the test results. To fulfill the limitations of this method, other alternatives, including biological and imaging analysis methods, also became commonly reported. The comparison of the different diagnosis tests allowed to understand the importance and potential of combining different techniques, not only to improve diagnosis but also for a further understanding of the virus, the disease, and their implications in humans.

## 1. Introduction

Acute respiratory syndrome coronavirus 2 (SARS-CoV-2), which causes the COVID-19 disease, appeared in December 2019 in China [1] and rapidly spread around the world, being declared by the World Health Organization as a global pandemic in March 2020 [2]. SARS-CoV-2 is an enveloped, positive-sense, and single-stranded Ribonucleic acid (RNA) beta-coronavirus, capable of infecting animals and humans. Regarding its biochemical constitution, the SARS-CoV-2 genome contains around 30,000 nucleotides, encoding structured proteins such as spike (S), membrane (M), envelope (E), and hemagglutinin esterase (HE), and nucleocapsid (N) [3]. The two SARS viruses’ S proteins were shown to have similar and low nanomolar binding affinities to human angiotensin-converting enzyme 2 (ACE2), the host surface receptor that the viruses exploit to enter cells (Figure 1) [4].

This virus can lead to respiratory, gastrointestinal, and neurologic syndromes [3]. Particularly, COVID-19 early infections’ most common symptoms are fever, cough, and other respiratory issues [5]. Although these are the main indicators of the disease, it is important to notice that some patients do not report any disease symptoms but may still contribute to the transmission of the virus to other human hosts [3]. 

The transmission of SARS-CoV-2 among humans occurs, mainly, via close contact with an infected individual that produces respiratory droplets and aerosols while coughing or sneezing, within a range up to 2 m [6]. Therefore, the scientific community recognized transmission by airborne particles as the primary route of infection, although contaminated surfaces or objects can also cause the virus to spread [6]. The main form of control and avoiding the virus transmission has been the use of personal protection equipment and social distance, as the general population has been oriented to use masks as a mechanical barrier to prevent droplet dispersion.

COVID-19 clinical diagnosis has been mainly based on signs and symptoms evaluation and confirmed by nucleic acid amplification tests (NAAT), for example, RT-PCR (Reverse Transcription Polymerase Chain Reaction) of nasopharyngeal or oropharyngeal swabs [7]. PCR-based methods are simple, highly sensitive, and highly specific and, therefore, they are routinely and reliably capable of detecting coronavirus infection in patients. These assays, widely used to amplify minimum quantities of deoxyribonucleic acid (DNA), start with the conversion of the coronavirus ribonucleic acid (RNA) into complementary DNA by reverse transcription. Subsequently, PCR is performed, and the resultant amplification of DNA is subjected to specific detection through different analytical methods. RT-PCR is a gold-standard method to detect most coronaviruses, including SARS-CoV-2. However, this method has some disadvantages, such as requiring expensive specialized equipment and highly trained analysts and technicians. Furthermore, PCR requires up to 4–8 h to process the samples and additional 1–3 days to report results, also having a high false-negative rate [8].

NAAT includes loop-mediated isothermal amplification (LAMP), multiple cross displacement amplification (MCDA), recombinase-aided amplification (RPA), CHAnicking and extension amplification reaction (NEAR), and Clustered Regularly Interspaced Short Palindromic Repeat associated (Cas) proteins (CRISPR–CasN)-based assays [9]. Serological tests can be performed as lateral flow immunochromatographic lateral flow assay (ILFA) [10], lateral flow immunochromatographic strip (LFICS) [11], chemiluminescence immunoassay (CLIA) [12], and enzyme-linked immunosorbent assay (ELISA) [13] based on antibody detection. Now, there have been some studies and commercial antigen detection kits for SARS-CoV-2 [14]. Although the molecular assay is the most common method to diagnose COVID-19, other diagnosis methods have been widely reported: chest computed tomography (CT) scan combined with the evaluation of clinical symptoms [15], potential electrochemical (EC) biosensors [16], field-effect transistor (FET)-based biosensors [17], surface plasmon resonance (SPR)-based biosensors [18] and artificial intelligence methods [19]. Figure 2 represents some of the tests. In order to diagnose various disorders, CT is a frequently employed auxiliary detection technology. The diagnosis of COVID-19 can be made on the basis of changes to lungs imaging brought on by SARS-CoV-2 infection [20].

Particularly, CT scans have several limitations for COVID-19 diagnosis, since they do not allow the identification of specific viruses, and many clinics and laboratories do not have access to proper equipment [19]. Regarding ELISA, this assay is based on the optical measurement of labeled fluorescent markers and, although ELISA has been used for the detection of SARS-CoV-2, the assay takes at least a few hours and requires specific spectral analyzers [21]. Although RNA detection based on RT-qPCR and antibody detection based on ELISA and LFICS have been developed, both methods have certain practical limitations. Concerning biosensors, they have the potential to be alternative tools since they can provide fast, accurate, sensitive early detection, especially smartphone-driven biosensors [18]. The advantages of EC biosensing assays are low cost, simplicity, easily miniaturization and mass manufacture. Additionally, they feature a rapid test result and can have the potential for being point-of-care, but their mass production for distribution is limited because it is a new technique for which the mass manufacturing process is in a process of development [19]. 

SARS-CoV-2 is highly contagious and, to date, although vaccines were approved and disseminated, there is no effective treatment for the disease. Due to the global pandemic state and the progressive human-to-human transmission, it became essential to develop preventive methods and COVID-19 diagnosis approaches while searching for novel treatments. As the research community started looking for solutions to diagnose COVID-19 faster and more efficiently, several approaches were developed to improve RT-PCR, and biosensors. Imaging analysis was used in machine learning algorithms to analyze CT and X-ray images on a large scale [22]. Since mass testing remains imperative, the present systematic review aims to identify the new advances in COVID-19 diagnosis that were achieved during 2021.

## 2. Methods

### 2.1. Search Strategy and Selection Criteria

The aim of this systematic review was to identify new solutions to diagnose COVID-19 and to overcome the disadvantages of the PCR gold standard tests. Thus, this review intended to analyze different innovative approaches developed and under development for COVID-19 diagnosis, compare them, and observe their advantages and disadvantages. 

The present systematic literature review was conducted according to the Preferred Reporting Items for Systematic Reviews and Meta-Analyses (PRISMA) guidelines. An electronic comprehensive search on ScienceDirect (SD), Scopus, and PubMed (PM) databases was performed. From database inception up to 21 January 2022, studies that included different types of tests for COVID-19, besides the real-time RT-PCR diagnosis, were searched. The search strategy was established by combining several keywords and the use of AND/OR Boolean operators. The relevant studies resulting from the database search were manually analyzed to identify other potential studies to be included. The exclusion criteria were: reviews, comments, overviews, case reports, viewpoints, and perspectives. Additionally, tests with low effectiveness and/or ambiguity of data were also excluded. Studies not written in the English language were also excluded. Figure 3 presents the PRISMA flow diagram of the conducted study. 

### 2.2. Data Collection

The database and additional manual searches provided 157 results. After removing duplicates, 78 studies were considered. First, titles and abstracts were screened. All abstracts were read and those that did not fit the purpose of this systematic review were excluded. According to the defined exclusion criteria, full texts were reviewed and, finally, data were extracted. Extracted data were: test design, experimental settings, quantitative outcomes and reported study limitations, as well as other relevant comments.

### 2.3. Outcomes

In the current systematic review, the outcomes of interest were the inclusion of a set of advantages and disadvantages/limitations of each diagnosis method. It was noticed that all the studies included in the review reported tests in few subjects, since COVID-19 is a very recent disease and, therefore, there were not a lot of samples available for research testing. Additionally, some of the studies also presented quantitative results such as sensibility and sensitivity. However, due to the low number of considered samples (around 100 samples per study), those parameters were not taken into consideration.

### 2.4. Data Analysis

All records were extracted to Mendeley software, where duplicated documentation was removed and manually checked. The titles of all documentation were searched by 3 authors, each one focused on one of the following categories of search: RT-PCR, Biosensors, and Imagology-Computational Methods. Sequentially, all abstracts and full texts were examined and evaluated by the same authors. In the occurrence of ambiguity, a consensus on the evaluation was reached by having an additional element.

## 3. Results

A total of 55 studies met all the eligibility criteria and were identified to be included in this review. Of those, 50 studies reported specific methods for diagnosis of COVID-19 and 5 studies were comparative.

Amongst the 50 studies, 38 were based on biological tests, and 12 reported computational techniques on imagological images. Some of the studies tested have their assays on clinical samples containing the SARS-CoV-2 gene, while others only reported tests in artificial samples. All the selected studies present the tests of the developed methods both on subjects infected with SARS-CoV-2 and controls (non-infected subjects). Table 1 presents the methods reported in the literature for COVID-19 detection using biological analysis assays, comprising the principle of operation, advantages, and limitations of all methods.

Table 2 presents the methods reported in the literature for COVID-19 detection based on imaging analysis, detailing the implemented machine learning algorithms.

## 4. Discussion

Due to the high variety of methods on which the COVID-19 diagnosis is based, particularly those exploiting the reaction chain of human antibodies, it would be interesting to perform a comparison between all the inherent characteristics of those studies, in order to properly discuss their advantages and limitations. Particularly, the sensitivity and specificity of the assays would be a relevant topic to address. However, since not all presented studies reported such information, and those who reported it were based on a small number of tested samples, it was not possible to faithfully compare those specifications. Additionally, another important parameter to assess the quality of the methods would be the cost-effectiveness per assay, which could not be defined or inferred due to the lack of well-founded literature, i.e., only one article reported that data [16]. Therefore, the comparison expressed here is sustained only by the main advantages, disadvantages, and reliability of the methods to be applied.

### 4.1. Nucleic Acid Amplification Tests

All the selected studies demonstrated that there is an urgent need to develop better methods for the diagnosis of COVID-19. RT-PCR, the most used technique, allows the processing of DNA/RNA to search for specific genes, and it is commonly used to diagnose a variety of viruses (for example, Grapevine virus T [57] and Zaire ebolavirus [58]). Particularly, in the case of SARS-CoV-2, the genes that are normally used for its detection are the N, S, E, M, ORF1a, and ORF1b genes [59]. The target gene regions, primer, and probe sequences used in different RT-PCR setups for SARS-CoV-2 detection are described by Yuce et al., 2021 [60]. As soon as the genomic sequence of SARS-CoV-2 was discovered, it was possible to perform RT-PCR on the discovered sequence, becoming the gold-standard method for COVID-19 diagnosis [61], assuring high accuracy.

RT-PCR is recommended as the most sensitive NAAT method [29,31,32]. However, most studies report RT-PCR as an assay that cannot fulfill the urgent requirements of the COVID-19 pandemic, as its technology relies on expensive and sophisticated equipment and reagents (with all the logistics complications that arise from their high prices), and it can only be performed by qualified people working in a laboratory specified to handle pathogens, it is also time-consuming and it may lead to some false-negative results [33,62]. Based on the classic and gold-standard RT-PCR technique, different studies reported new valuable methods, improved in comparison with the standard RT-PCR assays (as the rRT-PCR, with lower reagents and time consumption [28]), that would not need any new equipment or facilities in the existent medical units [33,49]. Also, digital PCR emerged as an improvement of RT-PCR. For example, Suo et al., 2020, optimized a droplet digital PCR used for the detection of SARS-CoV-2, which showed that the limit of detection is significantly lower than that of RT-PCR [63].

In addition, various isothermal techniques were performed. Isothermal techniques are a powerful tool that do not require expensive thermocycling or professional skills [64]. These isothermal amplification assays are more sensitive and independent of a heat cycler, making them better suited to the development of quick, high-throughput, and low-cost assays. LAMP is the most well-established isothermal amplification method [48], with LAMP assays for SARS-CoV-2 detection being the most widely used isothermal amplification method [25,44]. Although LAMP has many advantages, any aerosol produced can lead to false positives. LAMP can be combined with colorimetric [65], reverse transcription (RT) [66], and others [9]. Other tests can use this technology as DNA nano-scaffold based on SARS-CoV-2 RNA triggered isothermal amplification [26].

Given their high specificity, sensitivity, simplicity, and repeatability, CRISPR-based nucleic acid detection approaches have recently demonstrated significant potential in the development of next-generation molecular diagnostic technology and have been used to diagnose COVID-19 [67]. The CRISPR-CasN-based assay is an effective gene-editing technique, and it can be analyzed with a fluorescent reader or lateral flow strips [30] or in a system based on lateral flow. For example, Broughton et al., 2020, described a CRISPR-Cas12- based lateral flow to detect SARS-COV-2, it was shown to be a rapid and easy-to-implement method [67]. There have been several Cas-protein-based assays such as CRISPR-Cas12 [67], CRISPR-Cas3 [68], and CRISPR-Cas13 [69]. Before the detection of CRISP- R–CasN-based assays, an isothermal amplification step was frequently introduced, for example, Joung et al.,2020, described a system namely SHERLOCK based on RPA and Cas13 [69]. However, CRISPR-based assay requires further testing.

Nucleic acid aptamers are short, single-stranded DNA (ssDNA) or RNA molecules that are selected for binding to a specific target [70]. Recently, aptamers have begun to be applied to detect SARS-COV-2 [43]; however, few methods have been described until now. Song et al., 2020, described an aptamer to target the receptor-binding domain of the SARS-CoV-2 spike glycoprotein [71]. Aptamer-based detection is more flexible, less costly, more stable, and easier to produce than antibody-based assays [72]. Nevertheless, it still needs further investigations and optimizations.

### 4.2. Serological Tests

Alternatively, different research teams developed serological assays to detect COVID-19. These methods test a clinical sample, combined with specific antigens, to detect SARS-CoV-2 antibodies in the human subject and it can contribute to epidemiological investigations of COVID-19. The goal of this technology is to perform a qualitative or semi-quantitative evaluation of the antibodies using different techniques, such as ELISA, which consists of the antigen protein immobilized on the surface of microplate wells, binds to the target antibody [73]. CLIA combines chemiluminescence techniques with immunochemical reactions [74], and lateral flow immunoassays (LFIA) [75]. Currently, and to the best of the authors’ knowledge, the most promissory test seems to be LFIA, because it has the advantages of decreased technical requirement, affordability, lower sampling, and specimen preparation risk, higher detection sensitivity and specificity, and it can deliver results in 15 min [76].

Serological tests can be divided into antigen-based tests [37,40] and antibody-based tests [34,35,39,46]. Antigen-based diagnostics detect protein fragments on or within the virus. This type of testing can detect active infections within 15 min compared to hours with RT-PCR [77]. The widely available SARS-CoV-2 antigen kits employ two methods: (1) the ICT assay, which uses colloid gold conjugated antibodies to produce visible colored bands to indicate positivity, and (2) the FIA, which uses an automated immunofluorescence reader to provide results [78]. An antibody is a protein produced by the immune system in response to an antigen. Antibody-based diagnostic measures the presence/concentration of IgG and IgM levels in the blood/serum/plasma samples to determine if the body is fighting with a pathogen, for example, a contagious virus [79]. ELISA, LFIA, and CLIA are widely used in the detection of anti-SARS-CoV-2 antibodies [73]. These methods are not as specific as the tests recognizing RNA sequences in the virus.

This systematic review includes 3 articles that compare and analyze different serological tests for COVID-19 diagnosis [76,80]. While some of the tests were performed with only a specific antibody (IgG/IgM/IgA), others are performed with a combination of two of them. For each of the compared tests, their specificity and sensibility were presented. The main outcome of these works was the reporting that the results of serological immunoassays highly depend on the course of the COVID-19 disease, since the number of antibodies varies at different stages of infection [73,76,80]. Contrarily to RT-PCR, serological tests are less time-consuming, have a lower risk associated with the manipulation of specimens, and need fewer technical requirements. Serological tests can be useful as a complement to RT-PCR, especially in asymptomatic individuals, allowing the improvement of the epidemiological studies and the clinical diagnosis of COVID-19.

### 4.3. Biosensors

In addition to RT-PCR and serological testing, there is a lot of interest in developing new COVID-19 biosensors that are fast, reliable, and sensitive [81]. Some biosensors have been developed to achieve SARS-COV-2 detection. For example, some covid biosensors that have been developed were COVID-19 biosensors based on surface nucleoproteins that attach to the ACE-2 receptor [37], gold-nanoparticle biosensors [41], biosensors to detect antigen [42], FET [21], and ROS biosensors [27].

Biosensors based on FET technology are very promising for COVID-19 diagnosis due to their advantages, namely fast and ultra-sensible response [21,24]. Detection based on ROS levels is also promising, since these biosensors assure easy, fast, and cost-effective detection, with high sensitivity [27]. However, the large-scale production of these biosensors (based both on FET technology and ROS levels) and their distribution in medical units is yet a drawback that disables the possibility of currently using them. The electrochemical biosensor is another interesting form of biosensor for highly sensitive point-of-care SARS-CoV-2 detection [47]. In diluted human serum samples, Djaileb et al., 2020, based on surface plasmon resonance (SPR) using gold chips modified by viral nucleocapsid proteins, detected antibodies in 15 min with nanomolar sensitivity [82]. Biosensors have various advantages, including high sensitivity and specificity, low analysis costs, quick execution time, the low limit of detection, and the ability to construct tiny platforms that can be utilized directly at the point of care, but more research and testing are still required [83].

### 4.4. Imaging

Based on Table 2, it is possible to verify that there are several studies reporting machine learning algorithms in complement to radiological imaging techniques for COVID-19 diagnosis. It is possible to observe, from the reported studies, that there is a considerable preference for chest X-rays over CT scans for COVID-19 diagnosis. There are two main reasons for that [84]: (1) X-ray machines have a higher availability in the hospital environment, with lower inherent economical cost than CT scan machines; (2) X-ray scans involve lower ionizing radiations than CT scans. Besides the physical techniques usually applied in the acquisition of chest imaging, research teams worldwide have studied artificial intelligence [85], such as machine learning algorithms and deep learning to analyze those acquired images [86].

Machine learning algorithms, based on neural network architectures, such as CNN, combined with advanced artificial intelligence techniques embedded in radiological imaging, are useful for accurate COVID-19 detection or as an auxiliary tool for distinguishing COVID-19 from pneumonia [87]. Due to the higher accuracy and sensitivity of CNN and CNN-based model architectures, these architectures are applied more often than deep neural network (DNN) architectures [88]. By applying machine learning algorithms and optimization methods, it is possible to significantly diminish the processing time of a radiological test, highly reducing the time interval needed to acquire a COVID-19 diagnosis [89]. 

Many studies have described CT manifestations in COVID-19 patients; however, they have some differences because CT findings are strongly related to the stage of infection after symptoms begin [90]. Additionally, through pre-processing or pre-training methods, it is also possible to enhance the methods, improving their capacity to distinguish COVID-19 from pneumonia [91]. These methods also allow for a remote diagnosis, which avoids physical contact between the radiologist and the infected patient. However, these methods carry the necessity for large databases of medical images for pre-training and pre-processing, as well as an intrinsic cost for software programs [92]. Nevertheless, radiological tests, including CT imaging, are believed to be the assistive diagnostic method in which the presence of consolidation lesions in CT images indicates the onset of the COVID-19 disease [93].

### 4.5. Microfluidic Approach

Microfluid systems provide a platform for many diagnostic tests, including RT-PCR, RT-LAMP, nested-PCR, nucleic acid hybridization, ELISA, among others [94]. Microfluidic devices are made up of interconnected miniaturized compartments that can perform multiple experimental tasks, either individually or in parallel, in an integrated way. Usually, there are two distinct types of microfluidic devices, namely, paper-based and channel-based. The channel-based is manufactured by four main methods which are 3D-printing, molding, laminate, and nanofabrication. This type of microfluidic devices requires channels to create a reservoir for the integration of reagents. The paper-based microfluidic device is made of a series of nitrocellulose fibers or hydrophilic cellulose which guide liquid in paper by absorption [94]. Before the manufacturing process, these devices can be previously analyzed and developed using numerical simulations [95] to improve the performance of the sensor. The well-known advantages of the microfluidic systems, including the need for small volumes of samples, portability, and fast detection, are gaining increasing popularity as a tool to help to improve the detection and diagnosis of several diseases, such as malaria and diabetes [96,97,98,99,100] and also to evaluate the potential of novel therapies [101,102,103,104]. Additional microfluidic devices have been used commercially to detect and diagnose COVID-19. For example, during the second wave of COVID-19 in Italy, a microfluidic antigen was used in emergency rooms. The test showed high sensitivity, proving the potential of microfluidics as a tool for COVID-19 point-of-care tests [105]. The closest systems that use technology based on several microfluidic phenomena are the COVID-19 rapid test kits [9].

The use of microfluiddic point-of-care systems in the serological test is one of the research avenues being investigated for the diagnosis of COVID-19. Small sample volume, miniaturization, portability, multiplexed analysis, quick detection, and signal amplification techniques are advantages of microfluidic systems. They can also increase the sensitivity of analyte detection [106]. Additionally, new technologies have been developed to enhance the transfer rate of antigen and accelerate the reaction processes [107].

Recently, some researchers, like Torrente et al. [108], reported new approaches in which is possible to determine simultaneously both the viral and serologic status of an individual. They reported a novel multiplexed electrochemical platform ultra-rapid detection of COVID-19. Within physiologically relevant ranges, that platform quantitatively detects COVID-19-specific biomarkers in blood and saliva, including SARS-CoV-2 nucleocapsid protein (NP), specific immunoglobulins (Igs) against SARS-CoV-2 spike protein (S1) (S1-IgM and S1-IgG), and CRP [108].

Organs-on-a-chip have also been used in the research of alternative therapies for SARS-CoV-2 [109]. For instance, Si et al. used a human-airway-on-a-chip to investigate the antimalarial drug amodiaquine as a powerful inhibitor of infection with SARS-CoV-2 [110]. The research team modeled a human bronchial airway epithelium and pulmonary endothelium infected with pseudotyped severe acute respiratory syndrome coronavirus [110,111].

## 5. Conclusions

With the exponential growth of the pandemic situation caused by the coronavirus, a fast and effective method is needed to diagnose COVID-19, allowing mass testing of the population in a short period for greater control of the transmission. This systematic review presented the methods and assays, reported in 2021, for COVID-19 diagnosis, focusing on both biological and image analysis methodologies. The advantages, limitations, and main characteristics of each method were reported. Basically, PCR-based techniques are the gold standard for a reliable diagnosis but require highly trained personnel, and they are time-consuming. Diagnoses methods based on biosensors seem a viable option for a fast and sensitive response. However, this technology still needs to be produced on a large scale. Computational techniques in conjunction with the previous ones are very useful as a complementary diagnosis tool, but their processing takes a long time. In addition, while a plethora of methods and techniques are already available for an accurate COVID-19 diagnosis that can be individually tested, or as a complement to other methods, the limitations of the current methods and the instability of the SARS-CoV-2 coronavirus enhance the necessity of new developments in this field. Particularly, as SARS-CoV-2 variations are found, new improvements in the detection are required in the search for faster, more accurate, more sensitive, and simpler tests that are able to detect the virus as soon as possible and limit its transmission between humans.

## Figures and Tables

**Figure 1 micromachines-13-01349-f001:**
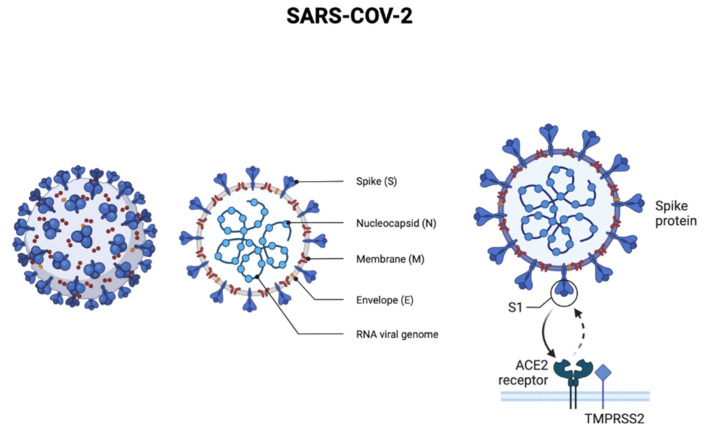
Sars-COV-2 virus representation.

**Figure 2 micromachines-13-01349-f002:**
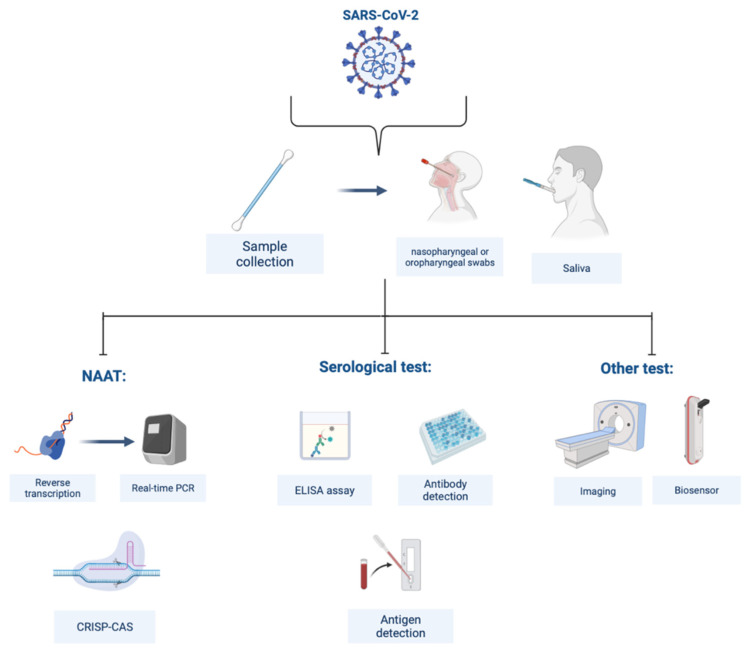
Sars-COV-2 virus diagnosis methods.

**Figure 3 micromachines-13-01349-f003:**
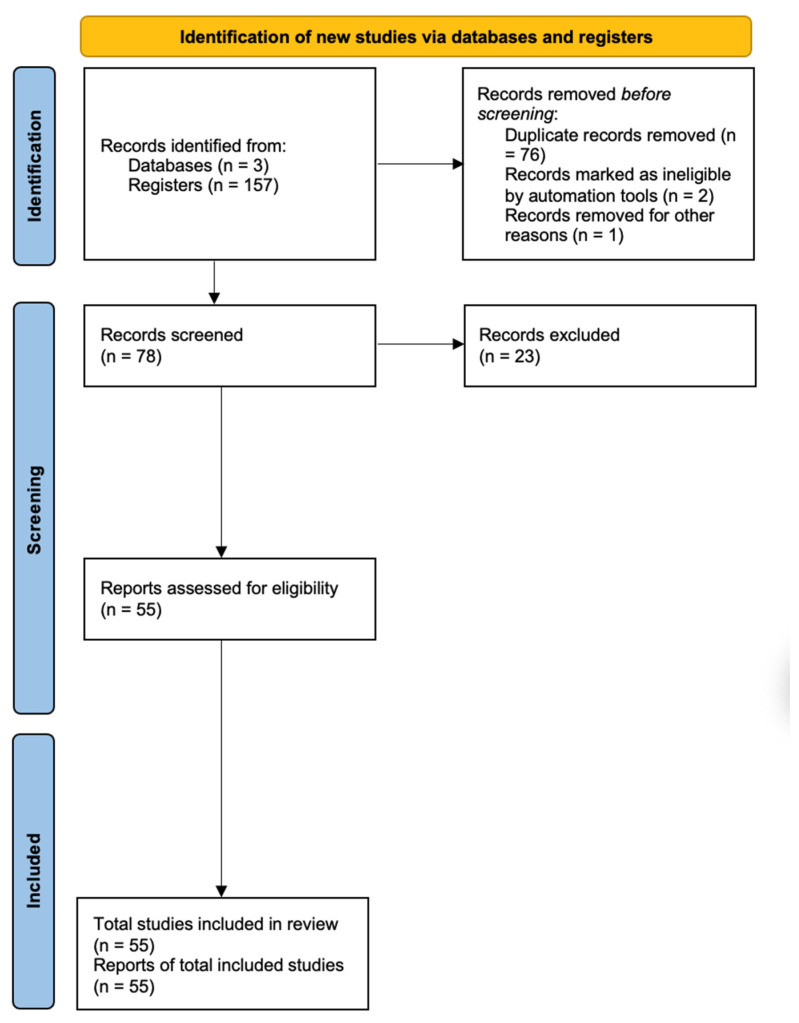
PRISMA flow diagram of search strategy conducted in the systematic review.

**Table 1 micromachines-13-01349-t001:** Methods reported in the literature for COVID-19 detection through biological analysis.

Detection Method	Principle of Operation	Advantages	Disadvantages/Limitation	Authors
Dual staining assay (Immunohistochemistry-IHC/in situ hybridization-ISH)	Reaction with S and N antibodies	Very precise; Useful for studying the pathogenesis of SARS-CoV-2	No quantitative results; Was not tested on clinical samples, only on Formalin-Fixed Paraffin-Embedded (FFPE) pellets	J. Liu et al. [23]
Graphene-based FET biosensor	Electric response of SARS-CoV-2 coupling S antibody	No cross-reaction with Middle East respiratory syndrome coronavirus (MERS-CoV); Highly sensitive and instantaneous measurement; Low-noise detection; Clinical samples do not need preparations/pre-processing	Needs novel materials for a more accurate detection	G. Seo et al. [24]
RT-LAMP	Auto-cycling strand displacement DNA synthesis using Orf1lab and S antibodies as target genes	Faster than RT-PCR assay; No cross-reactivity with other respiratory pathogens; Easy to handle; Does not require skilled personnel or specialized instruments; Results are easy to read	Few complete genomes available on databases; Mutation occurring with the spread of the virus	C. Yan et al. [25]
DNA nanoscaffold hybrid chain reaction (DNHCR)	The presence of SARS-CoV-2 triggers a cascade reaction along the DNA nanoscaffold, lighting up the structure (detected by fluorescence)	High signal gain; Short reaction time; High specificity; Room temperature response; Cost-effectiveness; Readily available reagents	Output of the fluorescence signal requires the use of specialized equipment	J. Jiao et al. [26]
Multiplex RT-LAMP coupled with nanoparticle-based lateral flow biosensor (LFB) assay-mRT-LAMP-LFB	LAMP amplification, reverse transcription, and multiplex analysis, allowing detection of orf1lab and N antibodies at the same time	Easy-to-use; Simple and objective; Less error-prone; Avoids the requirement of complex processes, special reagents and expensive instruments	RNA templates are sensitive to degradation by inadequate sample handling, post-mortem processes, or storage; Small number of clinical samples; Not evaluated on other clinical samples (e.g., sputum, blood, urine)	X. Zhu et al. [16]
COVID-19 associated ROS diagnosis (CRD)	Reactive oxygen species (ROS) released from infected cells would react with a working electrode covered by functionalized multi-wall carbon nanotubes, releasing electric charges that are posteriorly measured	Extremely rapid method; Non-invasive	Some false-negatives	Z. S. Miripour et al. [27]
Multiplex real-time RT-PCR (rRT-PCR)	Simultaneous detection of N and E gene	High sensitivity; Reduced reagents, costs and time required	Poor reproducibility; Maximum signal intensities were low	T. Ishige et al. [28]
Simplexa™ direct assay (RT-PCR)	Targeting of E and RdRp genes	Fast; Easy-to-use; Does not require extra laboratory equipment; Low train required; No cross-reactivity with other viruses	Small number of samples which can be tested in a run	L. Bordi et al. [29]
RT-PCR CRISPR-Cas12a	RT-PCR is used to amplify target regions from viral RNA and the resulting amplicons are transferred to the gRNA/Cas12a-based Clustered Regularly Interspaced Short Palindromic Repeats (CRISPR) system for fluorescence detection	Sensitive and robust; Readily available equipment	Small number of samples; Does not give quantitative results	Z. Huang et al. [30]
RT-PCR	E and RdRo detection	Very sensitive; No cross-reactivity with other viruses	Weak initial reactivity	V. M. Corman et al. [31]
RT-PCR	Fully automatic PCR platform for detection of E gene	No cross-reactivity with other viruses; Allows a large number of patients to be screened in a reasonable timeframe	Was not evaluated with clinical samples; Results have to be confirmed with an independent PCR	S. Pfefferle et al. [32]
rRT-PCR	Viral load detected in saliva samples	Low risk of transmission at the collection; Less invasive	Low number of samples; Qualitative results	L. Azzi et al. [33]
Serological Immunochromatographic (IC) assay	Immunochromatography strip assay for detection of IgM and IgC antibodies	Ready-to-use and time-saving; High detection capacity; Blood collection less risky than nasal swab samples	Assay carried out without specificity analysis; Qualitative results	Y. Pan et al. [34]
Quotient MosaiQ™ (microarray-based assay)	Automated detection of antibodies directed to the spike protein	High specificity and clinical sensitivity; Rapid throughput of samples; No cross-reactivity with other viruses	Weak repeatability and reproducibility; Did not include positive samples for other coronaviruses	C. Martinaud et al. [35]
BioFire^®^ Respiratory Panel 2.1 (RT-PCR)	Nucleic acid amplification platform for detection of M and S genes	Detects low levels of viral RNA; Allows the simultaneous differentiation between viruses; Easy to use	Non-specified	H. M. Creager et al. [36]
ACE2-based LFIA	Detect SARS-CoV-2 S1 protein using LFIA with a matched pair consisting of ACE2 and an antibody.	Detect the S1 antigen of SARS-CoV-2	Tests performed only on two different corona-related spike antigens	Lee et al. [37]
Cell-based biosensors for the detection of the SARS CoV-2 spike S1 protein	Molecular Identification through Membrane Engineering	Portable, high throughput, and low-cost system	No clinical validation	Mavrikou et al. [38]
Detection of IgM Antibodies against the SARS-CoV-2 Virus via Colloidal Gold Nanoparticle-Based Lateral-Flow Assay	A colloidal gold nanoparticle-based lateral-flow assay to detect the IgM antibody against the SARS-CoV-2 virus through the indirect immunochromatography method	Low sample consumption	Satisfactory specificity. Weak comparison with PCR results;	Huang et al. [39]
Fluorescent immunochromatographic assay based on multilayer quantum dot nanobead	Two-channel fluorescent Immunochromatographic assay method for ultrasensitive and simultaneous detection of SARS-CoV-2/FluA in real biological samples	Simultaneous detection of SARS-CoV-2 antigen and influenza A virus	Missing information about the number of samples tested	Wang et al. [40]
Gold nanoparticle-based biosensor	Combined colorimetric and electrochemical biosensor to detect SARS-CoV-2 spike antigen	Does not require for sensor preparation and modification. Saliva samples;	Satisfactory Selectivity	Karakus et al. [41]
SARS-CoV-2-specific biosensor for antigen detection	Lateral flow immunoassay-based biosensor using single-chain variable fragment-crystallizable fragment (scFv-Fc) fusion antibodies.	Time-saving, good detection limit	Needs optimization. Satisfactory sensitive	Kim et al. [42]
Aptamer	Detection of SRAS-CoV2 N protein using DNA-based aptamers	Aptamers can be synthesized easily and the process is less expensive than antibody production.	Lack of serum samples	Chen et al., [43]
Point-of-care nucleic acid amplification test for diagnosis of active COVID-19	Based on the principle of LAMP	Easy to operate and does not require skilled personnel	Some false-negatives	Deng et al. [44]
Point-of-care testing for SARS-CoV-2 virus nucleic acid detection	Catalytic hairpin assembly reaction-based signal amplification system coupled with a lateral flow immuno-assay strip	Highly sensitive, Fast.	Limited number of clinical samples	Zou et al. [45]
Reverse transcription–enzymatic recombinase amplification	Detect the SARS-CoV-2 gene by applying reverse transcription–enzymatic recombinase amplification	No need of thermocyclers	Dual detection and single-copy sensitive	Xia e al. [13]
CLIA	Serological test for detecting SARS-CoV-2 specific IgA as well as IgM and IgG	Measure levels of the three types of antibodies in blood	Few cases of COVID-19 patients	Ma et al. [46]
EC impedance-based detector	Electrochemical detection of SARS-CoV-2 antibodies using a commercially available impedance sensing platform.	Rapid screening of patient samples, expanded serological surveys to assess anti-SARS-CoV-2 antibody levels in the community.	Further testing is needed to determine the limit of detection	Rashed et al. [47]
RT-LAMP	RT-LAMP method designed to target the nucleocapsid protein gene	High sensitivity and specificity, low cost.	False-positive single read-out and sensitivity to aerosol contaminants during assay manipulations	Baek et al. [48]

**Table 2 micromachines-13-01349-t002:** Methods reported in the literature for COVID-19 detection through image analysis.

Methodology	Architecture	Machine Learning Algorithms	Optimization	Pre-Processing/Pre-Training	Reference
X-ray	Convolutional neural network (CNN)	Support vector machines (SVM); Decision Tree (DT); k-nearest neighbors (KNN)	Bayesian algorithm	None	M. Nour et al. [49]
	CNN	ConvXNet	Stacking algorithm	X-ray images of COVID-19 and other cases of pneumonia	T. Mahmud et al. [50]
	CNN	nCOVnet	VGG-16	X-ray images of COVID-19 true positive patients	H. Panwar et al. [51]
	Xception (CNN based)	CoroNet (SVM based)	Depends on the availability of the training data	ImageNet	A. I. Khan et al. [52]
	Residual Exemplar Local Binary Pattern (ResExLBP) and ReliefF (CNN based)	DT; Linear discriminant (LD); Subspace discriminant (SD); SVM; KNN	Local Binary Pattern (LBP); 10-fold cross-validation;	Leave-One-Out Cross-Validation (LOOCV); 10-fold cross-validation holdout validation	T. Tuncer et al. [53]
	CNN	SVM	Stochastic gradient descent (SGD)	Fuzzy Colour Technique MobileNetV2 SqueezeNet	M. Toğaçar et al. [54]
CT	Multi-Scale CNN (MSCNN)	Multi-scale spatial pyramid (MSSP) decomposition	None	2D Images	T. Yan et al. [55]
	CNN	Enhanced KNN classifier (EKNN); Hybrid feature selection methodology (HFSM)	KNN optimization	Gray Level Co-occurrence Matrix (GLCM) COVID_CT	W. M. Shaban et al. [56]

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
