# Peer review of "Diagnosis Methods for COVID-19: A Systematic Review"

_micromachines, 2022, doi:10.3390/mi13081349_

Round 1

Reviewer 1 Report

This work presents a systematic review of diagnosis methods for COVID-19. This kind of work is opportune and timely. My decision is recommend this manuscript for publication on these transactions with  minor remarks for the authors making the amendements before submit the final versions.

General remarks:

(1) Since the begining of 2020, it can be found on literature several review papers in the same subject. On what does your manuscript add to or differentiate from these works? Include this information in the introduction.

(2) The authors wrote an interesting resume about the most common methods for diagnosing the COVID-19 on conclusions, indicating their advantages and disadvantages. However, it is not pointed out future, e.g., does these methods will improve their accuracy and reliability? Can you state which are the new trends?

(3) On page 4/line 109, where it is stated “but their mass production for distribution is limited [19].”, it was interesting to include a justification about this limited distribution.

(4) On page 4/lines 117 e 118, I can identify the missing of a word, because it is not linked with it us previously written.

(5) Review the number of this sentence on page 5: “The database and additional manual searches provided 65 results. After removing duplicates, 54 studies were considered” seems not being in accordance with the figure 3. Fix it in accordance.

(6) On page 13;line 341, the sentence “However, to the best of our knowledge, microfluidic devices have been used commercially to detect and diagnose COVID-19.” Is not very well described. Review this portion of the text.

Specific remarks:

(1) in the end of the sentence on page 2/line 64 it should be included a reference.

(2) after the word “biossensors” on page 3/line 106 it should be also included a reference.

(3) increase the letter size of the texts on figure 1.

(4)  on page 11/line 240 change from “Recently, aptamers star being applied…” to ““Recently, aptamers start being applied…””.

(5) the sentence on page 11/line 260 to 260 sounds strange and awkward, making it very difficult to understand, and must be rewritten.

(6) on page 12/line 304 intead showing a reference, it shows “ERROR REFERENCE SOURCE NOT FOUND” and must be fixed.

(7) on page 13/line 347, the compound word “Point-Of-Care” has been used throughout the body of text without capitalization, appearing in this position with the first letter capitalized. Place the same throughout the document.

Author Response

The authors would like to acknowledge the careful revision done by the Reviewer, as well as his/her precious suggestions and comments. Also, we would like to thank to the Editor for the opportunity that was given to us regarding the improvement of the original version of the manuscript (MS) and the further resubmission of the improved MS. Hence, we hope that this new version of the MS is worth publishing in the Micromachines.

Comments and Suggestions for Authors

This work presents a systematic review of diagnosis methods for COVID-19. This kind of work is opportune and timely. My decision is recommend this manuscript for publication on these transactions with minor remarks for the authors making the amendments before submit the final versions.

General remarks:

(1) Since the beginning of 2020, it can be found on literature several review papers in the same subject. On what does your manuscript add to or differentiate from these works? Include this information in the introduction.

Answer to the reviewer: The authors are grateful for the suggestion to improve the MS. We did the changes suggested in the introduction. The new information is highlighted in yellow color.

(2) The authors wrote an interesting resume about the most common methods for diagnosing the COVID-19 on conclusions, indicating their advantages and disadvantages. However, it is not pointed out whether in the future, e.g., these methods will improve their accuracy and reliability. Can you state which are the new trends?

Answer to the reviewer: The authors are grateful for the suggestion to improve the MS. We did the changes suggested in the conclusions. The new information is highlighted in yellow color.

(3) On page 4/line 109, where it is stated “but their mass production for distribution is limited [19].”, it was interesting to include a justification about this limited distribution.

Answer to the reviewer: Thank you very much for your suggestion. We add the justification for the limited distribution. Please, see the text on lines 110-111 (highlighted in yellow color).

(4) On page 4/lines 117 e 118, I can identify the missing of a word, because it is not linked with it us previously written.

Answer to the reviewer: Thanks for your remarque, we add a word to link the two sentences.

(5) Review the number of this sentence on page 5: “The database and additional manual searches provided 65 results. After removing duplicates, 54 studies were considered” seems not being in accordance with the figure 3. Fix it in accordance.

Answer to the reviewer: Thank you very much for calling attention to this mistake. We corrected it and highlighted it in yellow color on MS.

(6) On page 13;line 341, the sentence “However, to the best of our knowledge, microfluidic devices have been used commercially to detect and diagnose COVID-19.” Is not very well described. Review this portion of the text.

Answer to the reviewer: Thank you for your suggestion. We rewrite the sentence (which is highlighted on the MS).

Specific remarks:

(1) at the end of the sentence on page 2/line 64 it should be included a reference.

Answer to the reviewer: Thank you for your recommendation. We added a reference that confirms our statement.

(2) after the word “biossensors” on page 3/line 106 it should be also included a reference.

Answer to the reviewer: Thank you for your recommendation. We added a reference that confirms our statement.

(3) increase the letter size of the texts on figure 1.

Answer to the reviewer: Thanks, we increased the letter in figure 1.

(4)  on page 11/line 240 change from “Recently, aptamers star being applied…” to ““Recently, aptamers start being applied…””.

Answer to the reviewer: Thank you for your observation we corrected the sentence.

(5) the sentence on page 11/line 260 to 260 sounds strange and awkward, making it very difficult to understand, and must be rewritten.

Answer to the reviewer: Thanks, we have rewritten the sentence

(6) on page 12/line 304 instead of showing a reference, it shows “ERROR REFERENCE SOURCE NOT FOUND” and must be fixed.

Answer to the reviewer: Thank you, we correct this error.

(7) on page 13/line 347, the compound word “Point-Of-Care” has been used throughout the body of text without capitalization, appearing in this position with the first letter capitalized. Place the same throughout the document.

Answer to the reviewer: Thank you for calling our attention to this mistake, we corrected it and highlighted the correction on the MS.

Reviewer 2 Report

In this manuscript, the authors reported a systematic review about the COVID 19 diagnosis methods and tests. The advantages, limitations of each method were investigated and summarized. This work is useful for the community. However, I have several concerns before this manuscript can be accepted. Therefore, in its current form, revisions are needed.

1. Microfluidic method has been exploited widely for detecting COVID19. The authors should discuss more content about microfluidic devices, since it owns many unique advantages including low cost, short assay time, miniaturization, and small sample volume, showing great potential in the point of care detection of diseases.

Such as, 10.1002/rmv.2154, 10.1007/s10404-021-02490-3, 10.3390/mi13020207

2. To effectively mitigate the risks of COVID19 community spread, devices are needed to determine simultaneously both the viral and serologic status of an individual. For example, Gao et al reported a multiplexed, portable electrochemical platform for detection viral antigen nucleocapsid protein, IgM, and IgG antibodies, as well as the inflammatory biomarker C-reactive protein. (10.1016/j.matt.2020.09.027). Thus multiplexed detection of different biomarkers using different methods will be desirable. Could the authors add some discussion about multiplexed detection of stage-specific SARS-COV2 related biomarkers for monitoring an individual’s health status?

Author Response

The authors would like to acknowledge the careful revision done by the Reviewer, as well as his/her precious suggestions and comments. Also, we would like to thank the Editor for the opportunity that was given to us regarding the improvement of the original version of the manuscript (MS) and the further resubmission of the improved MS. Hence, we hope that this new version of the MS is worth publishing in the Micromachines.

Comments and Suggestions for Authors

In this manuscript, the authors reported a systematic review about the COVID 19 diagnosis methods and tests. The advantages, limitations of each method were investigated and summarized. This work is useful for the community. However, I have several concerns before this manuscript can be accepted. Therefore, in its current form, revisions are needed.

  1. Microfluidic method has been exploited widely for detecting COVID19. The authors should discuss more content about microfluidic devices, since it owns many unique advantages including low cost, short assay time, miniaturization, and small sample volume, showing great potential in the point of care detection of diseases.

Such as, 10.1002/rmv.2154, 10.1007/s10404-021-02490-3, 10.3390/mi13020207

Answer to the reviewer: The authors are grateful for the suggestion to improve the MS. We read the references that you suggested and discussed more content about microfluidic devices. The new information is highlighted in yellow color. We also add the references suggested (highlighted in MS)

  1. To effectively mitigate the risks of COVID19 community spread, devices are needed to determine simultaneously both the viral and serologic status of an individual. For example, Gao et al reported a multiplexed, portable electrochemical platform for detection viral antigen nucleocapsid protein, IgM, and IgG antibodies, as well as the inflammatory biomarker C-reactive protein. (10.1016/j.matt.2020.09.027). Thus, multiplexed detection of different biomarkers using different methods will be desirable. Could the authors add some discussion about multiplexed detection of stage-specific SARS-COV2 related biomarkers for monitoring an individual’s health status?

Answer to the reviewer: Thank you very much for your suggestion. We analyzed the paper that you referred e discuss the new methodology presented by Gao et al, we also show the advantages of the multiplexed detection system. All this discussion is highlighted in yellow color on MS. Naturally, the suggested reference is also on MS.

Round 2

Reviewer 2 Report

The authors have addressed all my concerns, so I would like to recommend the paper for publication.